# On the Variability of the Circulation and Water Mass Properties in the Eastern Levantine Sea between September 2016–August 2017

Elena Mauri [1,*], Lina Sitz [1], Riccardo Gerin [1], Pierre-Marie Poulain [1,2], Daniel Hayes [3] and Hezi Gildor [4]

[1] Istituto Nazionale di Oceanografia e di Geofisica Sperimentale, OGS, Borgo Grotta Gigante, 42/c, 34010 Sgonico, Italy
[2] Centre for Maritime Research and Experimentation, CMRE, 19126 La Spezia, Italy
[3] Oceanography Center, University of Cyprus, P.O. Box 20537, 1678 Nicosia, Cyprus
[4] The Hebrew University Institute of Earth Sciences, Edmond J. Safra Campus, Givat Ram, 91904 Jerusalem, Israel
* Correspondence: emauri@inogs.it; Tel.: +39-040-2140-203

**Abstract:** The surface circulation and the thermohaline properties of the water masses of the eastern Levantine Sea (Mediterranean Sea) were monitored with mobile autonomous systems (surface drifters and gliders) during the period September 2016–August 2017. The drifters provided data for more than a year and revealed complex circulation features at scales ranging from the basin scale to the sub-mesoscale. Three drifters were captured in a semi-permanent gyre (Cyprus Eddy) allowing a quantitative study of its kinematics. During the experiment, three gliders were operated, in two different periods: September to December 2016 and February to March 2017. The autonomous instruments crossed the prevailing sub-basin structures several times. The collected in-situ observations were analyzed and interpreted in concert with remote sensing products (sea surface temperature and altimetry). The evolution of some of the prevailing features confirmed the complexity of the circulation of the basin. The Cyprus Eddy is the most persistent anticyclone, moving its geographical position and sometimes merging with the North Shikmona Eddy in a bigger structure. The gliders sampled this wide anticyclonic feature revealing its vertical structure in the two different periods. In fall, in stratified conditions, a high salinity core is evident below the thermocline. The isopycnals are characterized by an upward bending over the high salinity lens and a downward bending below it, typical of an anticyclonic modewater eddy. In winter, the core disappears following the vertical mixing that, homogenizes the upper Cyprus Eddy water down to 300 m.

**Keywords:** Mediterranean Sea; drifters; sub-basin-scale eddies; gliders

## 1. Introduction

The Levantine Sea (LS) is a complex multiscale system [1–3]. The basin-scale mean circulation, mesoscale and sub-basin scale eddies interact in a non-linear way producing a highly variable current field [4]. Despite many studies focusing on the LS, the basin has not been extensively sampled due to its high complexity and variability and to logistical or political issues.

Studies of the LS surface current started over a hundred years ago based on hydrographic data [5–11], revealing a basin-wide cyclonic circulation and the most persistent sub-basin scale features. A more detailed and complex circulation became evident only after the analysis of longtime series of satellite measurements, such as the Sea Surface Temperature (SST) [12,13] and the Absolute Dynamic Topography (ADT) [4,14–16] and numerical simulations [2,17–19]. Starting in 2005, as part

of the Eddies and GYres Paths Tracking (EGITTO) [20] and North East MEDiterranean (NEMED) [4] projects, numerous drifters were deployed in the region, allowing to calculate pseudo-Eulerian velocity statistics for different time periods. In particular, the use of the ADT for the period 1993–2010 allowed to describe the inter-annual variability of the Eddy Kinetic Energy (EKE) of the prevailing mesoscale features in the LS. The basin surface circulation map resulting from this study shows an along-slope cyclonic coastal circulation named the Libyo-Egyptian Current [12,20,21] extending as a northward current along the Middle-East coast. The other dominant feature is a central eastward cross-basin meandering current named Mid-Mediterranean Jet (MMJ) between 24° E and the longitude of Cyprus [22–24], and a series of mesoscale features, including some eddies such as the Cyprus (CE), the Shikmona (ShE) and the Latakia Eddy (LE). The CE is described as a persistent anticyclonic eddy characterized by seasonal variability in shape, dimension and position with an average diameter of 250–300 km [4,25]. The ShE instead, represents a complex system, composed of several cyclonic and anticyclonic eddies off the Israeli coast, in which the positions, sizes and intensities vary markedly [4,26–28]. The LE can be present as a cyclone or an anticyclone and the change in rotation is induced by the interaction of the MMJ with the northward meandering coastal current [4].

The thermohaline structure of the eastern LS is well-defined in the warm season and it is characterized at surface by the Levantine Surface Water (LSW), with temperature values between 22 and 28 °C and salinity of 39 to 39.6 PSU. The Atlantic Water (AW) with temperature values of 18 to 22 °C and salinity between 38.6 and 39.2 PSU, is positioned below the LSW and it is advected from the western Mediterranean Sea. The Levantine Intermediate Water (LIW) formed when LSW cools down and sinks along isopycnals to intermediate depths (ca. 130 m < $z$ < 350 m), it presents typical values of 15 to 17.5 °C and 38.95 to 39.3 PSU [29]. Finally, the Levantine Deep Water (LDW) with its nearly constant values of temperature and salinity (13.8 °C and 38.7 PSU) is found below 750 m [30].

In the framework of the CINEL (CIrculation and water mass properties in the North Eastern Levantine) project drifters and gliders were operated in the eastern part of the LS for more than a year, starting in September 2016, to gain more insights on the variability of the physical and biochemical properties in the region and in particular, to study the major sub-basin scale and mesoscale eddies governing the dynamics of the eastern LS. The in situ observations provided by the drifters and gliders were used in concert with satellite products of SST and ADT to describe the spatial structure and the temporal evolution of the main eddies.

The use of satellite images in past studies to track mesoscale and sub-mesoscale features has been widely used. In the Mediterranean Sea, SST, chlorophyll and altimetry imagery were utilized in different studies [4,16,31–34], some of them the eddies were sampled by gliders [35–38].

In this study, we concentrate on the eastern LS, located between 30° E and 36° E, and 31° N and 37° N in the period between September 2016 and August 2017. The main focus of the analysis is on the detection and monitoring of strong sub-basin scale features and on their motion and evolution during the period of the experiment.

The paper is organized as follows. Information on the in-situ platforms (drifters and gliders) and the data they provided, as well as on the remotely sensed data (SST and ADT), is provided in Section 2. The methods applied to process all the data are also explained. In Section 3, the features highlighted by the SST anomaly and ADT images are interpreted in concert with the drifter tracks and the surface geostrophic currents computed from the satellite altimetry data. The vertical description of some sub-basin scale features using glider observations is also included in this section. Discussion and conclusions are found in Section 4.

## 2. Data and Methods

### 2.1. Glider Data

Three Seagliders (sg149 and sg150 of University of Cyprus—UCY; sg554 of National institute of Oceanography and Applied Geophysics—OGS) were operated in the eastern LS between 1 September 2016 and 16 March 2017 (See Table 1). The glider dataset includes five glider campaigns

(Figure 1), which covered two different seasons: fall 2016 and winter 2016–2017. During the first period, three glider campaigns were organized; the first one was performed in September 2016 (C1) and the other two (C2 and C3) were almost concomitant and covered the month of November and the beginning of December 2016. The winter sampling comprised two simultaneous campaigns (C4 and C5) starting at the beginning of February and ending in mid-March 2017.

**Table 1.** Dates of glider campaigns in the two periods, conventional name of the missions used in the paper and number of recorded profiles.

| Glider ID | Fall 2016 | Name | Casts | Winter 2016–2017 | Name | Casts |
|---|---|---|---|---|---|---|
| OC-UCY (sg150) | 1 September–17 October | C1 | 203 | | | |
| OGS (sg554) | 19 October–7 December | C2 | 333 | 10 February–16 March | C5 | 127 |
| OC-UCY (sg149) | 4 November–6 December | C3 | 143 | 10 February–12 March | C4 | 168 |

The OGS sg554 glider is equipped with a pumped CTD (GPCTD) while the UCY gliders have a regular un-pumped CTD. The un-pumped data were corrected for the thermal lag using Kongsberg routines. Temperature and salinity data were collected in the top 950 m of the water column in all the five campaigns. Oxygen concentration was recorded by an Aanderaa optode 4330 and 3835 (for the OGS and UCY glider, respectively) and set to record data down to 600 m depth. All the gliders were also fitted with a Wetlab FLNTU sensor to collect chlorophyll and turbidity data at two wavelengths (470 and 700 nm) down to 300, 400 m or 600 m while crossing specific structures. The glider traveled along a ~26° inclined path with respect to the water surface. During the up casts, the glider collected high vertical resolution data (about 0.1 Hz, corresponding to about one sample every 2 m). The top 20 m CTD data of the OGS campaigns are missing. This behavior is common in Seabird GPCTD and is due to the incomplete download of the GPCTD buffer when the glider reaches the surface (SeaBird, 2013). The horizontal speed of the instrument spanned between 0.7 and 1.4 km/h depending on the sea currents. A glider took about 4–5 h to conclude a 950 m dive, therefore, the horizontal resolution was about 4 km. The data were transmitted via satellite Iridium links at each surfacing. In total, 974 casts were collected during the five missions.

During fall 2016 (campaigns C2 and C3), the OGS and UCY gliders were operated simultaneously. On 3 December, they were close by and the two profiles (one per glider) recorded at that particular moment (separated by about 1.5 km and 1 h) were used to intercompare the CTD data. Temperature and salinity profiles showed a good agreement (offset of 0.02 °C and 0.01 PSU, RMSD of 0.07 °C and 0.01 PSU). Oxygen data were not inter-compared because the oxygen factory calibration for the UCY gliders resulted too old and a sensor degradation (shift over times) was suspected. Therefore, only the relative values of the oxygen concentration are used in this paper. Chlorophyll and turbidity data were corrected using the dark counts computed from deep dives performed at the beginning and at the end of each glider campaigns. After a first quality control to eliminate obvious spikes [39], the data were averaged in 2-m non-overlapped bins. In this paper, the temperature, salinity, density and dissolved oxygen data are described while other parameters are only occasionally considered.

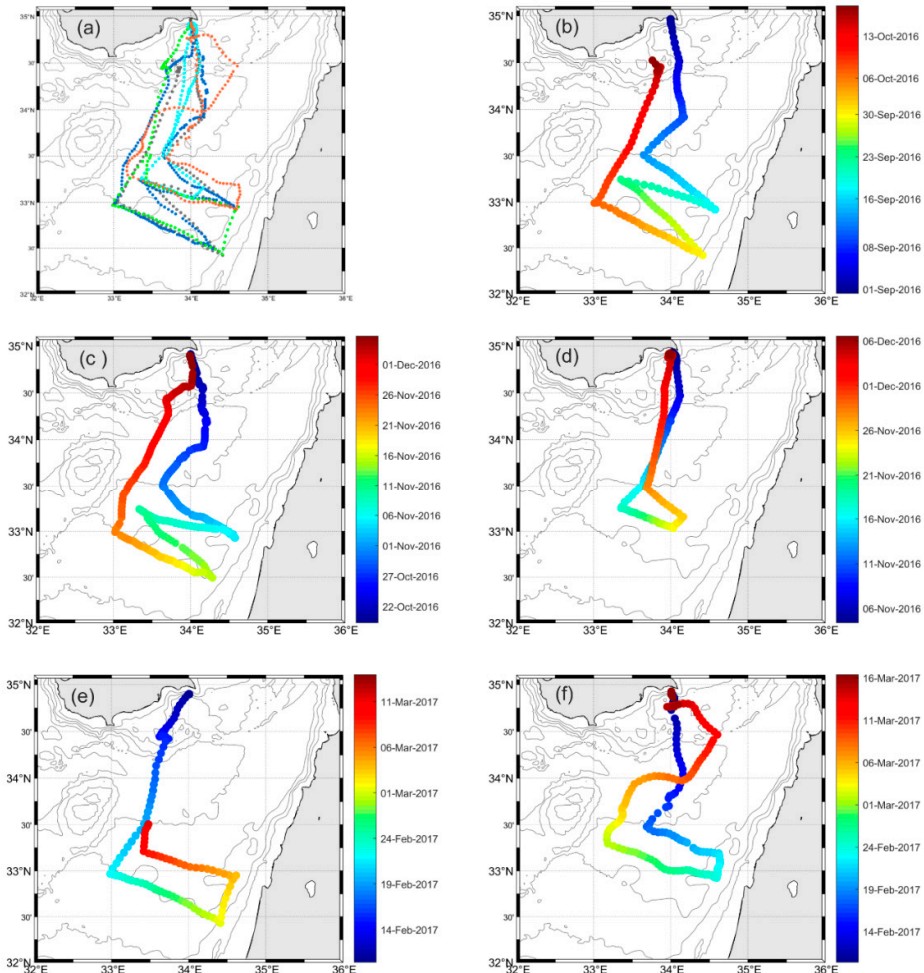

**Figure 1.** (**a**) Glider tracks during the CINEL experiment, (**b**) C1 mission performed by units SG150 glider during fall; (**c**) C2 by unit SG554 in fall, (**d**) C3 by unit SG149 in fall; (**e**) C4 by SG149 in winter; (**f**) C5 by SG554 in winter. Colors represent the evolution in time of each trajectory, with blue colors indicating the initial measurements and red colors showing the final observations. The actual dates for each mission are reported in Table 1.

## 2.2. Drifter Data

The CINEL drifter dataset includes the tracks provided by 16 drifters launched in 3 episodes: 4 drifters on 20–21 October 2016 along a meridional transect south of Cyprus and 8 drifters off the central coast of Israel on 7 February 2017 and 4 drifters again south of Cyprus (same positions as in October 2016) on 25 February 2017. One drifter stranded near Larnaca in Cyprus on 14 March 2017. It was recovered and redeployed south of Cyprus on 29 March. The drifters used during the experiment were the Surface Velocity Programme SVP drifter design [40] with a drogue centered at 15 m depth, equipped with a sensor for the SST and a tension sensor to monitor the drogue presence. They were manufactured by METOCEAN. Each drifter provides its location through the global positioning system (GPS) and transmits the data on land via Iridium satellite link. The drifter data were first edited from spikes and outliers [39], then the data of position, temperature, voltage and drogue presence were interpolated at 30-min uniform intervals using a kriging optimal interpolation method [41,42]. The velocities were then calculated as finite differences of the interpolated position. The interpolated positions were also subsampled at 2-h regular interval and low-pass filtered using a Hamming filter with a cut-off period at 36 h in order to remove high frequency current components

(tidal and inertial currents) and then sub-sampled every 6 h. A composite diagram with all the low-pass filtered drifter trajectories between 20 October 2016 and 31 August 2017 is shown in Figure 2.

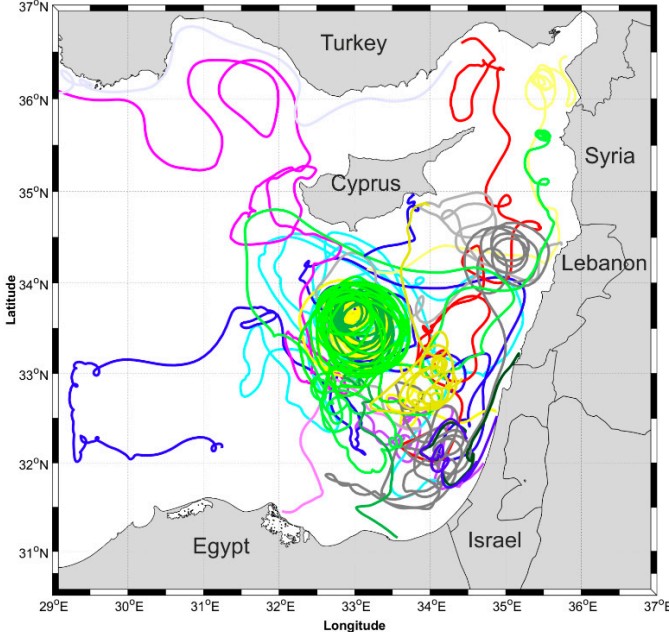

**Figure 2.** Tracks of drifters operated during the CINEL experiment from September 2016 to August 2017.

Drifter trajectory segments were superimposed on SST anomaly and ADT images to describe specific snapshots of surface circulation. A quantitative analysis of the drifter trajectories was also performed to characterize the CE, when some drifters were caught in its core for several months. To estimate the motion of the eddy center, loops in the drifter tracks were identified by considering the drifter positions bounded by two successive longitude maxima [43]. Then, the center of each loop was computed by averaging the longitudes and latitudes corresponding to each loop. A regression line was fitted through the center estimates to estimate the mean displacement of the eddy. The size and strength of the eddy were determined by considering the drifter speeds as a function of distance with respect to the eddy center.

### 2.3. Sea Surface Temperature

The satellite SST products considered in this study are daily gap-free maps (L4) at high (HR 0.0625°) spatial resolution over the Mediterranean Sea. These products are based on night-time images collected by the infrared sensors mounted on different satellite platforms and cover the Southern European Seas. Remotely-sensed L4 SST datasets are operationally produced by the Consiglio Nazionale delle Ricerche-Gruppo di Oceanografia da Satellite (CNR-GOS). The basic design and the main algorithms used are described in [44]. The products are distributed by the Copernicus programme (http://marine.copernicus.eu). In this study, we generated SST anomaly maps for the east LS (30–36° E and 31–37° N) with a constant colorscale (between −2 and +2 °C) in order to enhance the features present in the area. In particular, each image was generated after subtracting the mean SST of the area and dividing the result by the standard deviation.

### 2.4. Absolute Dynamic Topography

The altimetry data are processed by the DUACS multi-mission altimeter data processing system and provided by Copernicus. Satellite gridded Sea Level Anomaly (SLA) are computed with respect

to a twenty-year mean and is estimated by Optimal Interpolation [45], merging the measurement from all the available altimeter missions (Jason-3, Sentinel-3A, HY-2A, Saral/AltiKa, Cryosat-2, Jason-2, Jason-1, T/P, ENVISAT, GFO, ERS1/2) (see QUID document or http://duacs.cls.fr pages for processing details). The SLA data resolution is 0.125° by 0.125°; to derive the ADT, the SLA is added to the Synthetic Mean Dynamic Topography (SMDT). The geostrophic currents are then derived by finite differencing and normalizing the ADT.

## 3. Results

### 3.1. Surface Circulation and Sub-Basin Features

The analysis of the drifter trajectories (Figure 2), and the SST anomaly/ADT daily images confirm the eastern LS as a very dynamical area, characterized by a number of permanent eddies and other features that are intermittent, eventually disappearing or merging with other eddies, creating wider structures. The ADT averaged throughout the whole period of study (September 2016–August 2017; Figure 3) and the drifter tracks (Figure 2) show the CE as the most persistent feature present in the area, covering a broad area south of Cyprus. Other relevant anticyclones are the North Shikmona Eddy (NSE), which is the easternmost feature off the Lebanese coast and another eddy located more to the south in front of the Israeli coast, hereafter called E1. In between these two anticyclones, a cyclone is evident. It is referred to as the South Shikmona Eddy (SSE) hereafter. Finally, another two cyclones, evidenced in the mean ADT, can be seen: one south of the CE, named E2 and the second one east of Cyprus island called the Latakia Eddy (LE). The above-mentioned features with the exception of the CE, are not permanent throughout the year, they can disappear for short periods or move slightly their geographical position or sometimes change their shape merging with other eddies. A qualitative analysis of the most salient features during a 1-year period starting from September 2016 was performed using SST anomaly and satellite altimetry maps. To analyze the mesoscale circulation, geostrophic currents computed from the altimetry data of the same day were superimposed on the daily SST anomaly images. Drifter tracks were also integrated to study the dynamics of the area.

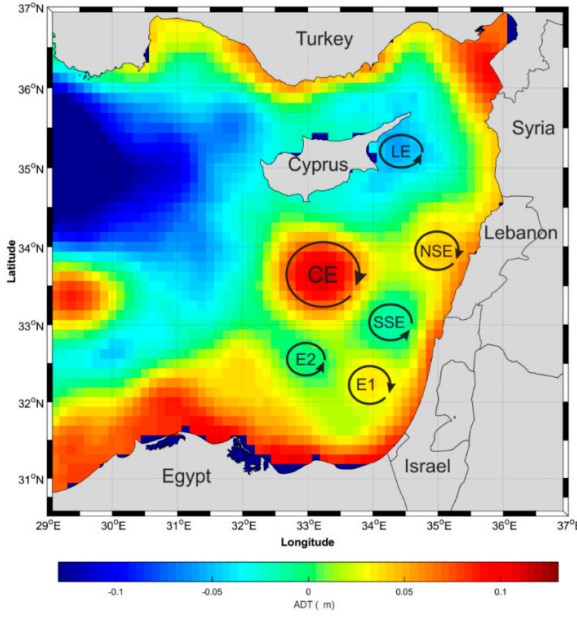

**Figure 3.** Mean Absolute Dynamic Topography (ADT) for the period September 2016 to August 2017, evidencing the Cyprus eddy (CE), the north Shikmona Eddy (NSE), the south Shikmona Eddy (SSE),

the anticyclone E1 in front of the Israeli coast, a cyclone south of CE called E2 and the Latakia Eddy (LE). The colorscale unit is m.

### 3.1.1. Qualitative Description

Mid Mediterranean Jet and Thermal Front between Cyprus and Syria

From the beginning of September 2016 a strong jet (Figure 4a), crossing from west to east the eastern LS south of Cyprus, was evidenced by low temperature and by strong geostrophic currents. The wind over the whole basin blew from the west starting from June 2016 and becoming stronger and more stable at the beginning of August and lasting until 25 September 2016 (https://winds.jpl.nasa.gov/missions/quikscat/). The jet, called from now on the Mid Mediterranean Jet (MMJ), bended toward Cyprus skimming the CE to the north, and proceeding toward the Syrian coast. Before reaching, the coast, it flowed to the north between two eddies: the LE to the north and the NSE to the south (Figure 4b). The cold water of the MMJ (1.5–2 °C lower than the surrounding water), intruded from the west, joining the coastal upwelled water along the southern coast of Cyprus. In early October, the cold MMJ faded away (Figure 4c) and the warmer waters of the CE, intruded to the north, confining the cold upwelled water to the southeastern part of Cyprus (Figure 4c–i). The cold water, together with the general cooling of the area northeast of Cyprus, generated a zonal thermal front (Figure 4e–i). From September to November 2016 (Figure 4a–f) the strong geostrophic currents flowing from south of Cyprus toward the Syrian coast, were mainly located south of the front (in warmer waters). Three of the four drifters deployed along the meridional section of the CE in October 2016, clearly highlighted the zonal jet associated with the front. The northern drifter was immediately captured by the jet (Figure 4d) and reached the Syrian coast after about a week, then it veered to the north (not shown). The drifter deployed in the CE more to the south, after being involved for 10 days in a small eddy southwest of CE, was first caught by the CE, then it moved to the north and entered in the jet (Figure 4e). On 21 November, it reached the Syrian coast. Another drifter reached this area on 25 November and then both instruments moved southward along the Lebanese coast (Figure 4f). The geostrophic current in front of the Lebanon/Syrian coast, computed by the altimetry, show for this period a current flowing against the drifter tracks (not shown). One drifter was caught in the SSE, while the other one proceeded to the south along the coast until 11 December when it changed direction and moved northward to reach the SSE, in agreement with the geostrophic currents (not shown).

The zonal thermal front northeast of Cyprus persisted until March 2017 changing its shape and orientation. After a first phase in September 2016 during which the front appears jagged and not well defined in the SST anomaly (Figure 4c), in October and November it became sharper with a surface temperature gradient of 1.5–2 °C (Figure 4f). Two gliders were operated in November and both crossed the front simultaneously a few km apart. During this period, the zonally oriented front persisted until 15 December when a cyclonic eddy located around 34° E, intruded in the warmer side bringing colder water to the south (not shown). The wide cold bulge expanded more than 1 degree of latitude to the south in 1 December (Figure 4g) and the intrusion became narrow due to the weakening of the cyclone by 11 January 2017 (Figure 4h). A small feature of cold water, which detached from the above-mentioned bulge, remained at 33.5° E 34.3° N in mid-January (Figure 4i) and was included in a small cyclonic eddy (probably the SSE). On 23 February, two cold intrusions of 10 km size propagated to the east along the front moving more than 40 km in 4 days (Figure 4j,k). In early March the jet decreased in intensity and more meanders and filaments developed. Some of them were also identified by the drifter tracks (Figure 4l).

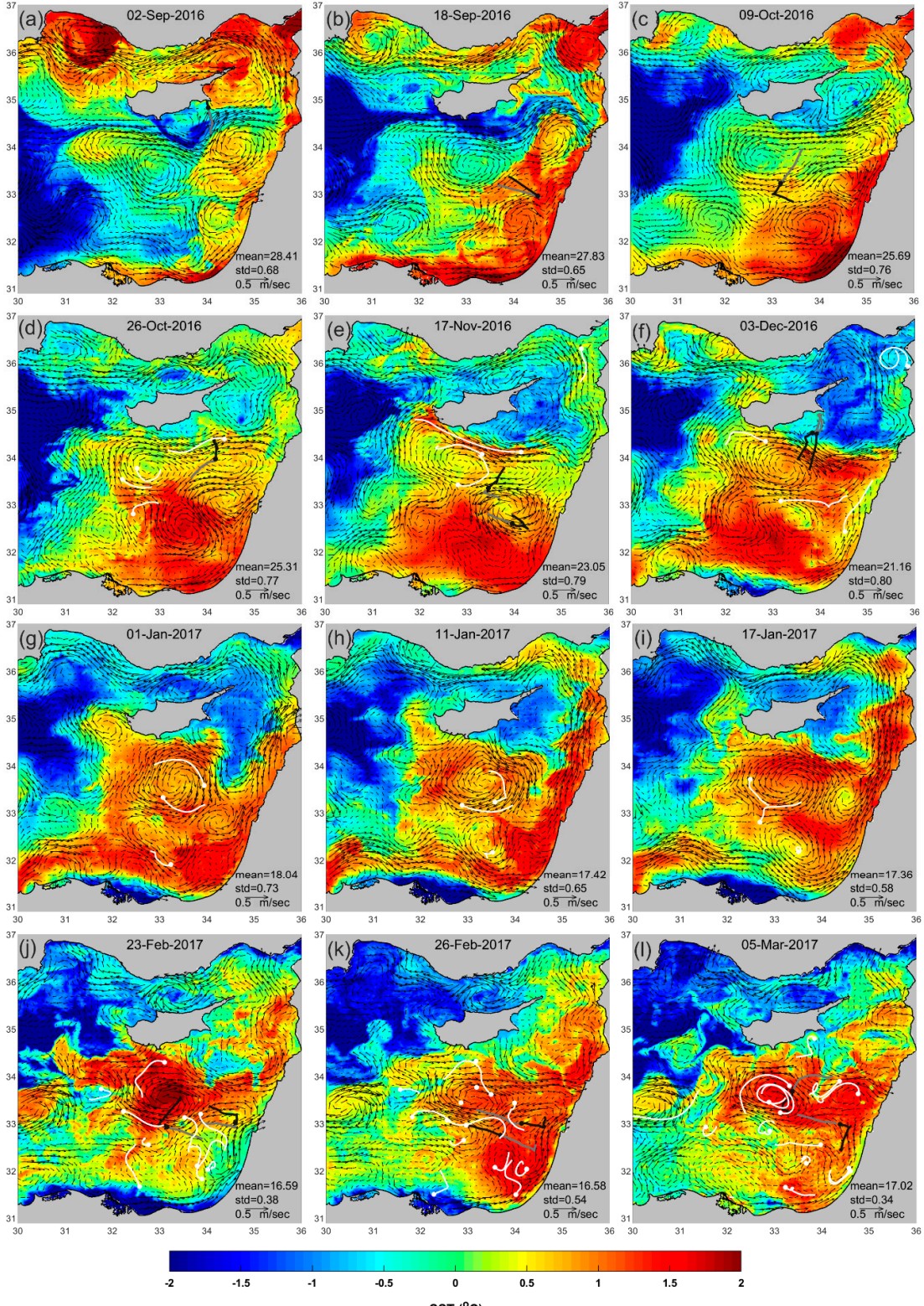

**Figure 4.** Geostrophic currents overlaid on the Sea Surface Temperature (SST) anomaly images, near the bottom the right corner of the sub-plots, mean, and standard deviation of the image are reported as well as the speed scale of the currents. Black (gray) lines indicate the glider track 4 days before (4 days after) the date reported on top of the image (black dot). White lines represent the drifter tracks 4 days before the reported date (white dot). Mid-Mediterranean Jet (MMJ) and the upwelling south

of Cyprus, marked by cold water (**a**); the fading of MMJ with instabilities along the edge (**b**); cold front south east of Cyprus (**c**); cold front highlighted by a drifter (**d**); a second drifter evidences the fast geostrophic current north of Cyprus (**e**); drifters proceeding south along the Lebanon coast (**f**); north intrusion in the front (**g**); evolution of the intrusion (**h**); creation of a separate cyclonic eddy (**i**); intrusions (**j**); evolution of the intrusions (**k**) and cold front weakening (**l**).

Cyprus Eddy and North Shikmona Eddy

Even though the CE prevailed throughout the period of study from September 2016 to the end of August 2017, its shape and position changed significantly with time as revealed by the analysis of the ADT maps. At the beginning of September, the CE and NSE were present as two separate anticyclones (Figure 5a), then, on 4 October, three small eddies developed (Figure 5b) and in a few days, a wide zonally-elongated anticyclonic structure formed (Figure 5c). This broad feature persisted (Figure 5d) until the beginning of December, when the two eddies eventually detached from each other (Figure 5e) and the CE moved back to its original location, while the NSE completely disappeared (Figure 5f).

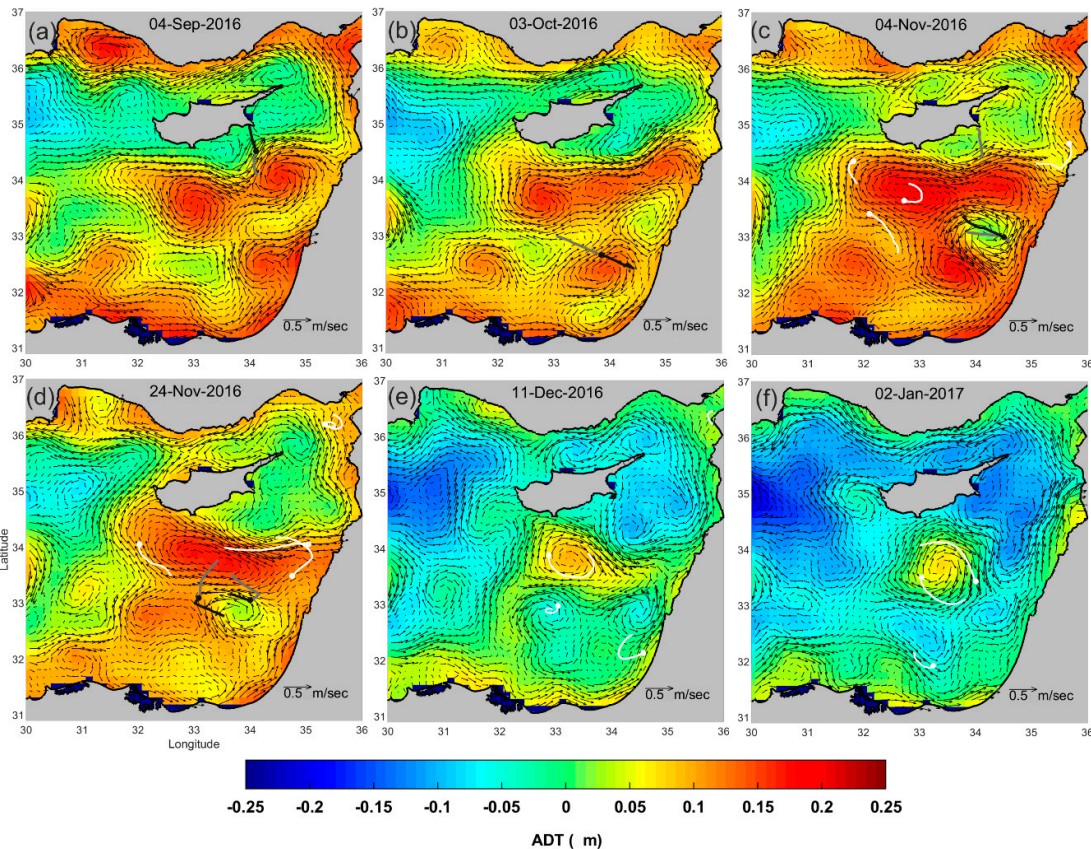

**Figure 5.** Selected maps of ADT with overlaid geostrophic currents with black, gray and white lines and dots as in Figure 4. CE and NSE evolution; anticyclones as separated eddies (**a**); presence of 3 small eddies in the same area (**b**); wide zonally-elongated anticyclonic structure (**c**); persistence of the CE-NSE anticyclone (**d**); weakening of the elongated anticyclone eddies (**e**) and disappearance of NSE (**f**).

To better show the time evolution of the two eddies and their geographical position, two Hovmoeller diagrams using as reference the altimetry data at 33.81° N (Figure 6a) and 34.2° E (Figure 6b) were produced. The latitude and longitude of the CE were selected based on the mean ADT for the period September 2016 to August 2017 (see Figure 3). As mentioned before, from October to December 2016 a large zonally-expanded anticyclone structure characterized the area. In November,

this feature covered zonally the broadest area from 32° E to 35° E (Figure 6a) and in the meridional direction from 33.4° N to 34.7° N (Figure 6b). Starting from December, the CE was evident again as a single anticyclone with a lower signature in ADT and a smaller extension of about one degree in both longitude and latitude.

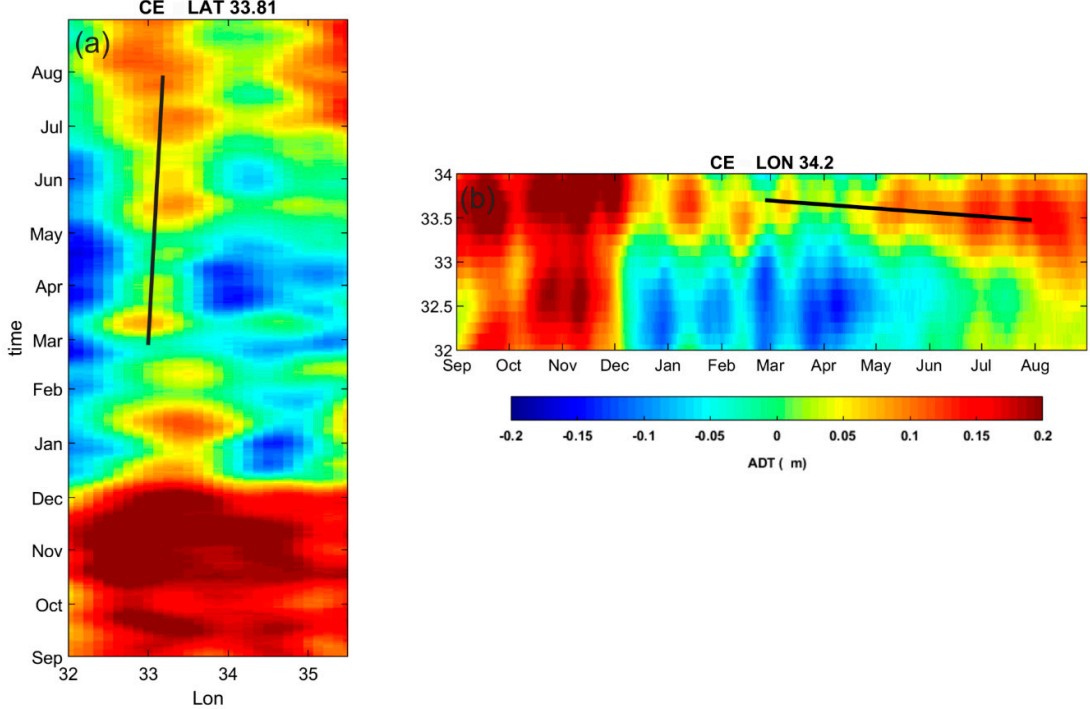

**Figure 6.** Hovmoeller diagram of ADT (**a**) along latitude 33.81° N and (**b**) longitude 33.19° E. Position of the CE determined from the annual ADT image. The black line shows the eddy displacement derived from the drifter data.

In September the gliders sampled the water column in the CE while in October-November 2016 they captured the CE-NSE merging event (refer to paragraph 3.2.).

In stratified conditions, from May until August 2017, the anticyclone was evidenced by a core of cold water (Figure 7). The area south of Cyprus was characterized by a strong upwelling with cold waters advected southward by the CE (Figure 7a–c). Starting from May, the eastern part of the basin becomes increasingly cooler. In July, cold filaments were advected toward the north, tracing the external edge of the CE (Figure 7b). By mid of August 2017, the cold western water was not entering as MMJ as in the previous year (Figure 7c). From June until August the NSE moved more than ½ degree north, despairing for 2 weeks in mid-July.

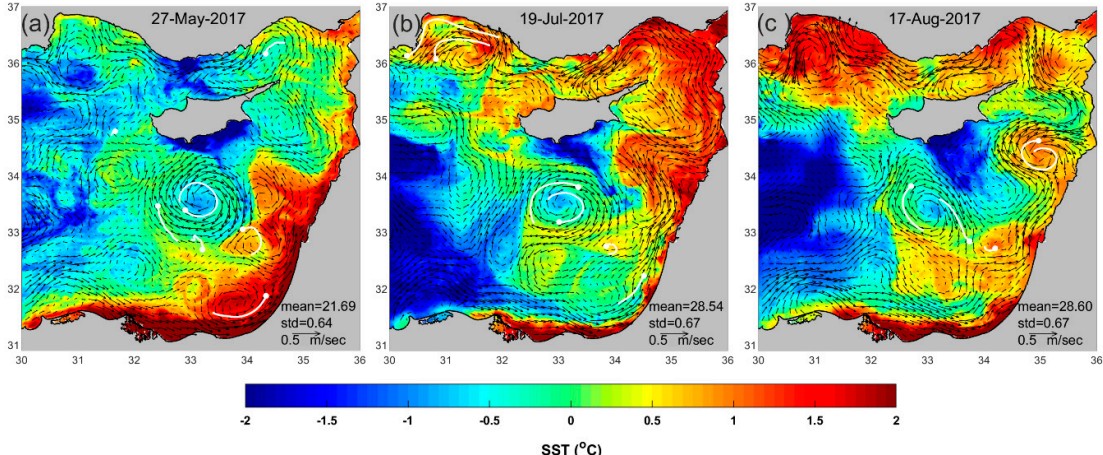

**Figure 7.** SST anomaly images with overlaid geostrophic current with white lines and dots as in Figure 4. CE marked by a cold core from May to August 2017; cold core and upwelled water south of Cyprus advected by the CE (**a**); CE cold core and a westward filament tracing the external edge of the CE (**b**); CE extending to the Cyprus coast, upwelling south of Cyprus, absence of the MMJ (**c**).

Upwelling off Israel and South Shikmona Eddy

In late November 2016, an event of upwelling took place along the Israeli coast and lasted for about 7 days. The phenomenon, induced by eastern wind blowing for 10 consecutive days over 3 m/sec, was visible from 26 November to 2 December 2016. The upwelled water showed a temperature decrease of 2 °C close to the coast. The SSE and other smaller eddies to the south probably advected the cold water in the open sea and the offshore flowing filaments maintained a difference in temperature while moving away from the coast (Figure 8). The two gliders sampled the area during the same period, but they did not capture the upwelling signature. The cold layer, advected away from the coast, probably affected only the top surface layer, which was not sampled by the gliders.

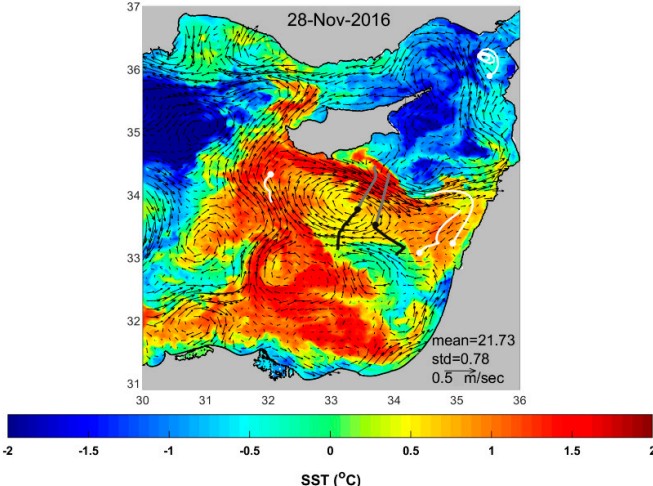

**Figure 8.** SST anomaly image with overlaid geostrophic currents with black, gray and white lines and dots as in Figure 4. Event of upwelling in front of the Israeli coast at the end of November 2017.

During fall and winter, the ADT maps show a distinct motion of the SSE. In order to have a better description of the 1-year eddy evolution Hovmoeller diagrams of the ADT were constructed

(Figure 9). The reference position (latitude 33.06° N–longitude 34.31° E) of the eddy was identified from the mean ADT map (Figure 3). In September during the first glider campaign the eddy was weakly detected in ADT, then from October to the end of November during the second and the third glider campaigns the eddy became well defined (Figure 5c) and moved southwestward (½ degree south and 1 degree west; Figure 9a,b). In early December, the SSE moved farther west and a new SSE appeared at the usual position. From mid-December to the beginning of May, the SSE alternated periods when the eddy was well defined and periods in which it merged with another cyclone intruded to the south; sometimes the south tongue was close to the Lebanon coast (Figure 10a), other times the presence of NSE pushed the intrusion farther offshore (Figure 10b). The SSE finally splits in two cyclones at the end of May 2017 as shown in Figure 10c. During the summer months the eddy was weak or disappeared for weeks while in August it moved ½ degree north (Figure 7c).

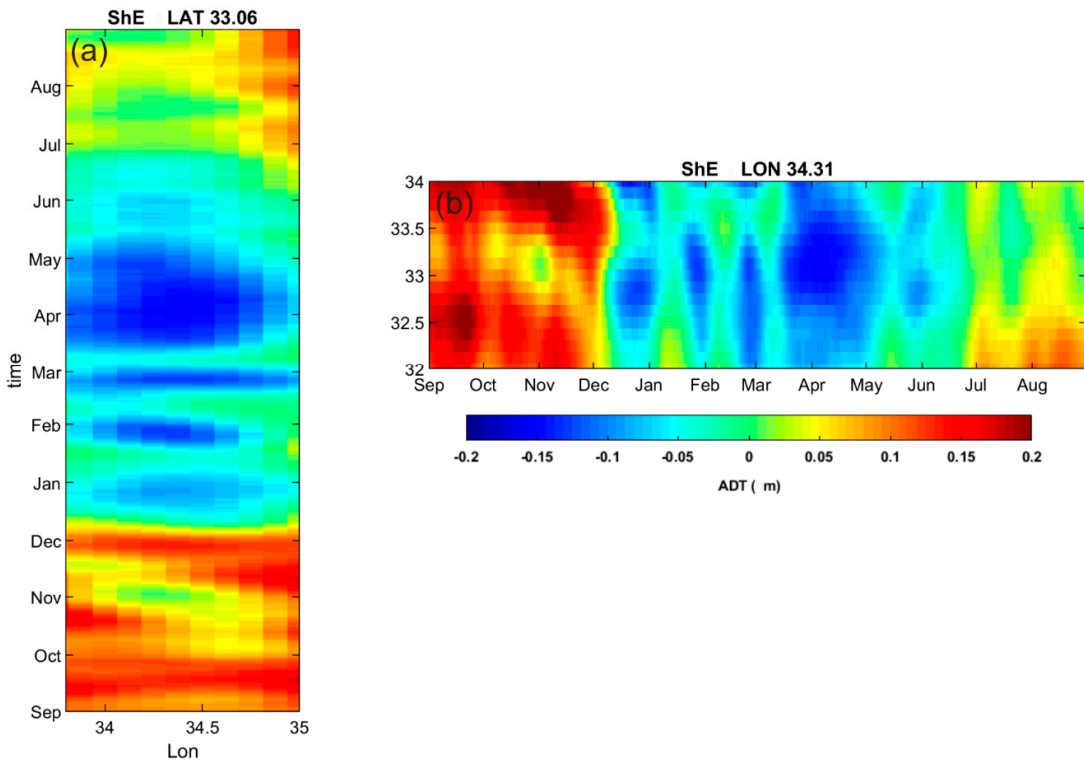

**Figure 9.** Hovmoeller diagram of ADT (**a**) along latitude 33.06° N and (**b**) longitude 34.31° E.

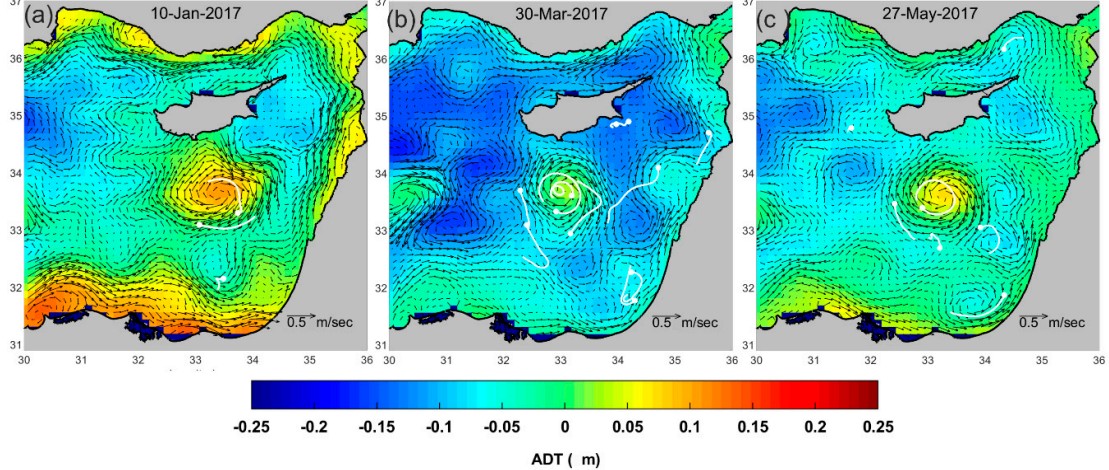

**Figure 10.** ADT with overlaid geostrophic currents with white lines and dots as in Figure 4. SSE merged with a cyclone intruded from the north: sometimes the intrusion was close to the Lebanon coast (**a**); the presence of NSE pushed the northern intrusion father offshore (**b**); SSE splits in two cyclones (**c**).

### 3.1.2. Quantitative Description of CE Using Drifter Data

Three drifters moved in and around the CE from February to May 2017, while two of them remained trapped in the eddy until August 2017. From the analysis of the data of the 3 drifters it was found that the eddy can be considered to be in quasi-solid body rotation up to a radius of 40–50 km, where the maximum drifter speed of 50 cm/s occurred [43]. In the 4 months period, the radius ranged between <10 km to about 100 km with some oscillations, due to the fat that the eddy might not be perfectly circular and also due to possible variations in rotation strength. From the estimation of the eddy center position, it was found that the anticyclone moved toward the southeast at a speed of about 150 m/day (see more details in [43]). This displacement is in good agreement with the ADT maps. The black line depicted using the coefficients computed by the drifter data follows the motion of the CE displayed by Hovmoeller diagrams (Figure 6).

### 3.2. Qualitative Vertical Description of Some Sub-Basin Features Using Glider Surveys

In Fall 2016, the three glider campaigns were designed to sample the vertical structure of some sub-basin scale features between September and the beginning of December. The data of the second campaign (see track in Figure 1c) were plotted as a function of time (Figure 11). The eastern LS area appears strongly influenced by the presence of sub-basin eddies and mesoscale features. The thermohaline structure throughout the mission, shows a strong vertical stratification, with the presence of warm and salty LSW near the surface down to 40 m (Figure 11a), which gradually deepens, cooling and freshening (Figure 11b). The temperature (salinity) parameter between 20 m to 40 m reaches a maximum value of 26.24 °C (39.57 PSU). By the end of the mission, the mixed layer has deepened to 90 m and the maximum temperature has decreased to 20.38 °C (39.36 PSU). The AW is found at depths just below the LSW, under the thermocline (at about 50 m in October and 100 m depth at the end of November), revealed as a local minimum in salinity (about 38.8 PSU; Figure 11b), which slowly disappeared as the season comes to the end. Furthermore, a 70 m thick layer with relative salinity maximum is observed between 100 and 400 m corresponding to the thermohaline characteristics of the LIW. Below that depth in the LDW, the salinity gradually decreases with depth (not shown).

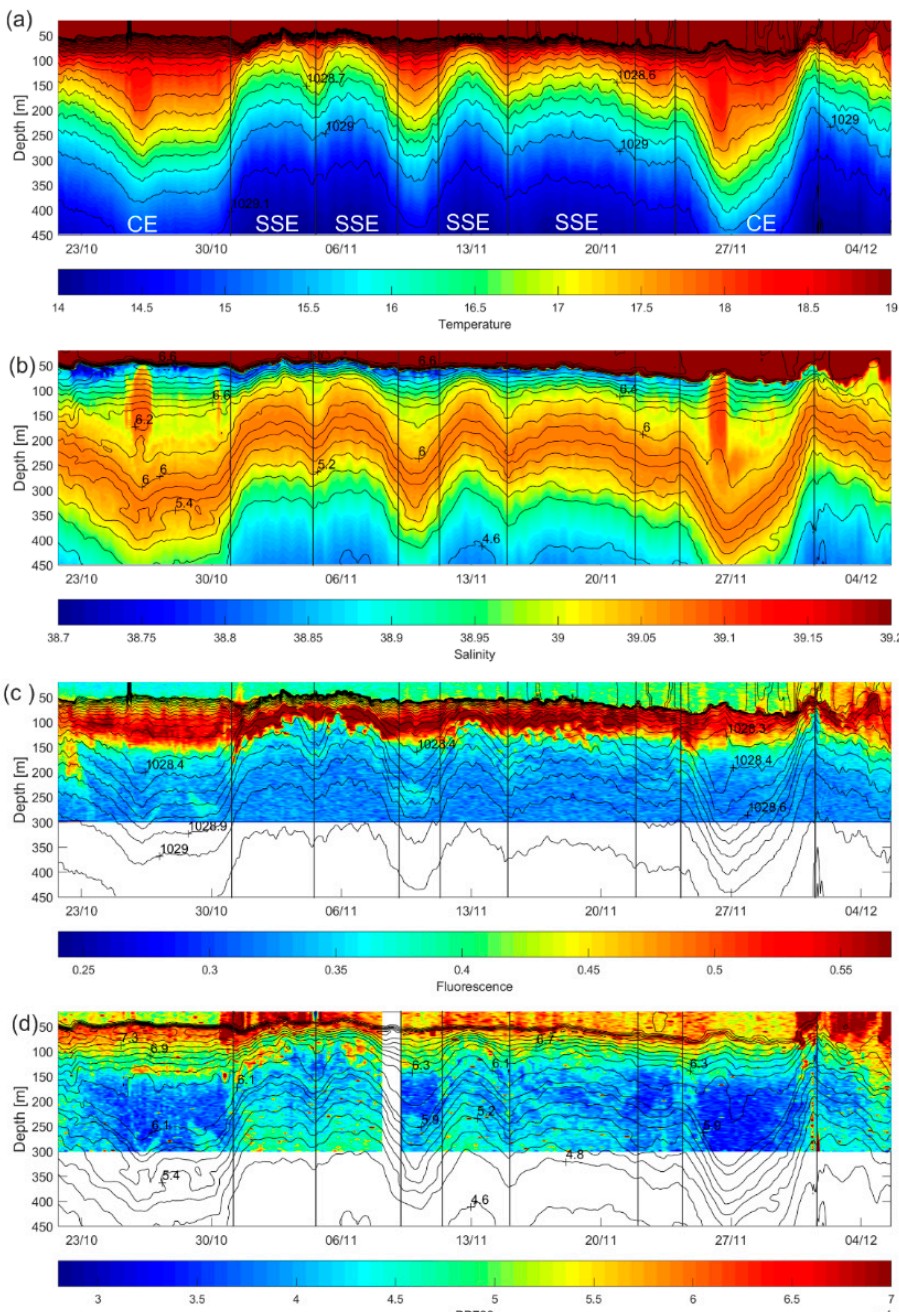

**Figure 11.** Glider data collected during the second campaign as a function of time and depth (20–450 m). Temperature field (°C) with isopycnal (intervals of 0.1 kg/m³) overlaid (**a**), salinity field with oxygen (intervals of 0.2 mL/L) overlaid (**b**), fluorescence field (µg/L) with isopycnal overlaid (**c**), backscattering at 700 (m⁻¹) with oxygen overlaid (**d**).

The contour plots of the second campaign (Figure 11) reveal the downward bending of the isolines of the water mass properties in correspondence to the CE anticyclone at the beginning and at the end of the mission. In between these anticyclones, a sequence of 4 transects, marked by individual upward domings, depicts different parts of the SSE area. The data after the second SSE transect (10–11 November) corresponds to the CE only partially sampled. After the fourth transect (22–25 November) data describe an anticyclone south of CE. The anticyclonic and cyclonic features are identified by acronyms and vertical black lines in Figure 11. The isopycnal curves overlaid on

the temperature field show a homogeneous top layer and a highly stratified layer in correspondence to the AW below the thermocline.

The contour lines, representing the oxygen concentration (overlaid in Figure 11b) show a subsurface maximum between 65–100 m, diminishing gradually along the water column, while the top layer is quite homogeneous. The oxygen stratification is evident throughout the mission with the exception of the CE, where the isolines between 150 m depth and the LIW show a less stratified field. Some mesoscale structures visible in temperature, salinity and oxygen are present inside the CE (see description in the following section). The chlorophyll concentration (Figure 11c) displays a Deep Chlorophyll Maximum (DCM) at around 50 and 150 m depth. Its spatial structure (thickness and patchiness) is strongly influenced by the presence of the eddies. In the anticyclone, the DCM is deeper than in the cyclone, where the chlorophyll concentration is slightly higher. Maximum values are around 0.8 µg/L while the minimum is around 0.2 µg/L. Some filaments detach from the DCM in particular along the edges of the anticyclone, reaching 200–250 m depth. The backscattering (Figure 11d) exhibits different vertical distributions in the anticyclones and cyclones. In the CE backscattering is relatively high between 50 and 150 m, while inside the eddy below 150 m it is extremely low. This feature is present not only in the two CE transects but also in the area where two anticyclones were partially sampled (10–11 and 22–25 November). The extremely low backscattering values in the CE (the absolute minimum of the mission) also corresponds to the region of lower stratification in oxygen concentration. In the deeper layer down to 600 m (not shown, because sampled in mission C3) the backscattering is low, but the recorded absolute minimum is still between 150 and 300 m depth. The cyclone exhibits higher values of backscattering in the upper layer in the first 2 crossings of the SSE, then when the winter mixing starts, the values are lower even in the upper part of the SSE. There are relatively higher backscattering values below 200 m in the SSE area.

The Cyprus Eddy and North Shikmona Eddy

During fall 2016, the area of the CE formation was sampled five times (see Table 2 for timing and other details). Since the gliders crossed the anticyclones along almost meridional transects (in both directions), the data were plotted in Figure 12 as a function of latitude for a better description and comparison. Each crossing transect was completed in 5 to 10 days from the beginning of October to the beginning of December 2016.

**Table 2.** Season (F—fall and W—winter), numbers assigned to the transect crossing the CE, mission name, period and approximate direction of sampling.

| Season | Transect | Mission | Glider ID | Date | Glider Direction |
|:------:|:--------:|:-------:|:---------:|:----:|:----------------:|
| F | 1 | C1 | sg150 | 07 October 2016–15 October 2016 | Northward |
| F | 2 | C2 | sg554 | 21 October 2016–30 October 2016 | Southward |
| F | 3 | C3 | sg149 | 7 November 2016–17 November 2016 | Southward |
| F | 4 | C2 | sg554 | 23 November 2016–02 December 2016 | Northward |
| F | 5 | C3 | sg149 | 27 November 2016–02 December 2016 | Northward |
| W | 6 | C4 | sg149 | 17 February 2017–21 February 2017 | Southward |
| W | 7 | C5 | sg554 | 02 March 2017–06 March 2017 | Northward |

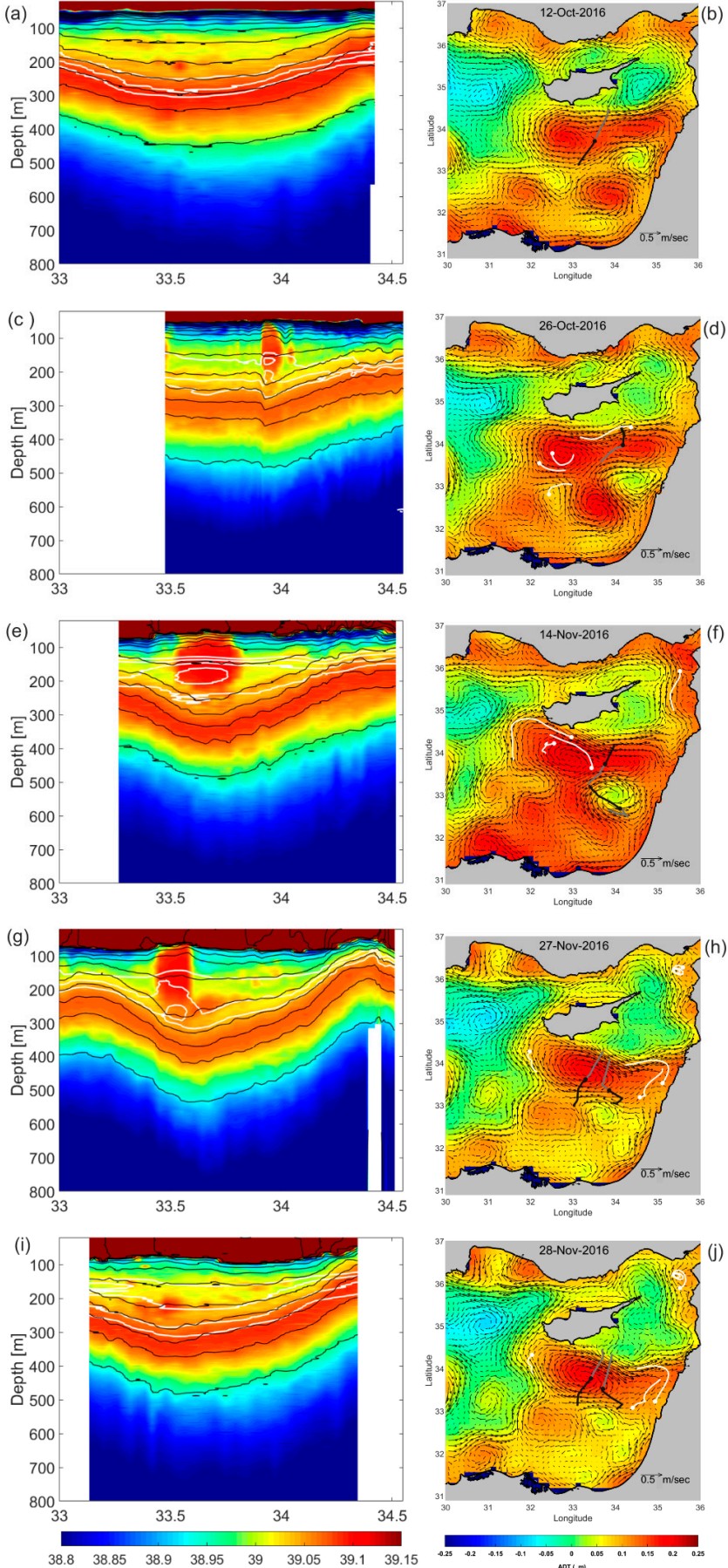

**Figure 12.** Fall salinity fields of the anticyclonic structures in the region of the CE (**left**). Transects are ordered chronologically and dates are reported in Table 2. Isopycnal (intervals of 0.1 kg/m³) and isolines of oxygen concentration (intervals of 0.2 mL/L) are respectively shown with black and white lines. On the right, the ADT field and geostrophic currents. Black, grey, white lines and dots as in Figure 4. Salinity sections depicted in panels (**e,g,i**) correspond to the northern, western and eastern glider tracks in panels (**f,h,j**), respectively.

All five salinity fields collected during fall are displayed in chronological order in Figure 12. The ADT fields, with the associated geostrophic currents and glider/drifter tracks overlaid, (Figure 12, right column), give additional information of the surface dynamics during the sampling interval. The dates of the selected ADT maps correspond to the times when the gliders sampled the center of the isolines deepening. The black dot on the glider track corresponds to its position on that date. From the analysis of the ADT the first transect (Figure 12b) crosses a zonally elongated anticyclone corresponding to the merging of the CE and NSE, described in paragraph 3.1.1. Transect 2 (Figure 12d), one week later, crosses the wide formed structure close to the NSE usual position. As time passes the anticyclone becomes larger with strong geostrophic currents as observed during the third glider crossing (Figure 12f). Approximately one week later (Figure 12g–j), the same structure is captured almost simultaneously along transects 4 and 5. The transect 4 evidences the eddy in its central part, while the transect 5 is relative to its easternmost portion.

On the left side of Figure 12 the salinity vertical structure with density (black curves) and oxygen (white curves) overlaid, describes the five sampled transects of the anticyclone in the CE area between October and the beginning of December 2016. The water masses of the CE are typical of the eastern LS but the thermohaline vertical structure is strongly influenced by the deepening of the isolines extending down to 950 m. It is noticeable that all the transects exhibit an asymmetry in all parameters that does not seem to be related to the sampling method, since it is present both in the transects sampled from north to south (Figure 12c,e) and in those sampled in the opposite direction (Figure 12a,g,i).

A high salinity subsurface core of different shape and size is present in all transects. The homogeneous salinity core or lens is around 39.1 PSU and mainly enclosed within an isohaline of 39.05 PSU. The vertical extension of the core in transects 2, 3, 4 is between 50 and 250 m and laterally from 33.4 to 34° E (more specifically 33.4–33.6° E in transect 2, 33.5–33.8° E in transect 3 and 33.9–34° E in transect 4). This core is marked also in temperature (not shown): the 18 and 18.5 °C isotherms are bended downward, surrounding the lower part of the lens (e.g., Figure 11a). The core in its upper part evidences an upward doming in salinity, density and oxygen concentration.

The isopycnals highlight a strong vertical stratification in density in the upper part of the core while the lower part is rather homogeneous. A minimum of oxygen is clearly visible in the lower part of the lens (white lines in Figure 12c,e,g). The high salinity core evident in transects 1 and 5 (Figure 12a,i) have smaller vertical extension spanning between 250 and 300 m and laterally from 33.3 to 33.5° E in transect 5, while it is barely visible in transect 1.

In the attempt to have a better perception of the high salinity core geographical position, the five glider pathways, crossing the CE, were overlaid on the bathymetry of the area (Figure 13). The dots correspond to the salinity at 150 m and they are color-coded according to the salinity values. The transects are numbered following a chronological order. During the first and fifth transect a small high salinity core is barely visible because the lens is located deeper (Figure 12a), while in transect 2, 3, 4 the core is well captured. Figure 13 shows that the salty core was sampled first to the west during transect 2, then it was captured ½ degree to the east in transect 3 and again ½ degree to east during transect 4. Transect 5 was almost concomitant to the fourth transect but did not capture the salty core at 150 m. High salinity is also evident in the northern part of the transects, where the bathymetry is shallower than in the southern part, but this is another feature different from the signature of the high salinity core.

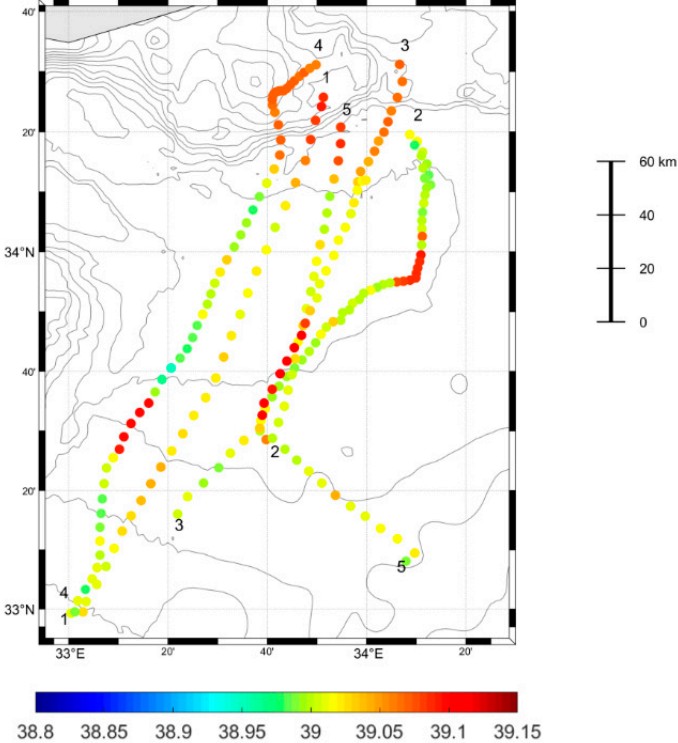

**Figure 13.** The fall 2016 glider pathways corresponding to the transects shown in Figure 12. The color-coded dots are the salinity sampled at 150 m depth. Transects are numbered chronologically according to Table 2.

In winter, the gliders crossed the CE twice (see transects 6 and 7 in Table 2) and only transect 6 is described as an example. The deepening of the water mass properties is evident. The salinity field is displayed in Figure 14a, while its geographical position is shown on the ADT image in Figure 14b. The CE appears like a body of homogeneous salty water (39.2 PSU) down to 400 m depth, included in the downward doming of the LIW. The temperature in the surface layer (not shown) is homogeneous only to 100 m depth in the northern edge and 200 m in the southern edge, confirmed by the isopycnals (black lines in Figure 14a). The AW is no longer visible as a local minimum in salinity, having been mixed with the layers below and above during the winter in the previous weeks. The LIW is also no longer identifiable as a relative maximum in salinity but it still conserves the thermohaline properties found during fall (Figure 15). In the winter structures, both chlorophyll and turbidity fields (not shown) are vertically homogeneous reaching 300 m depth.

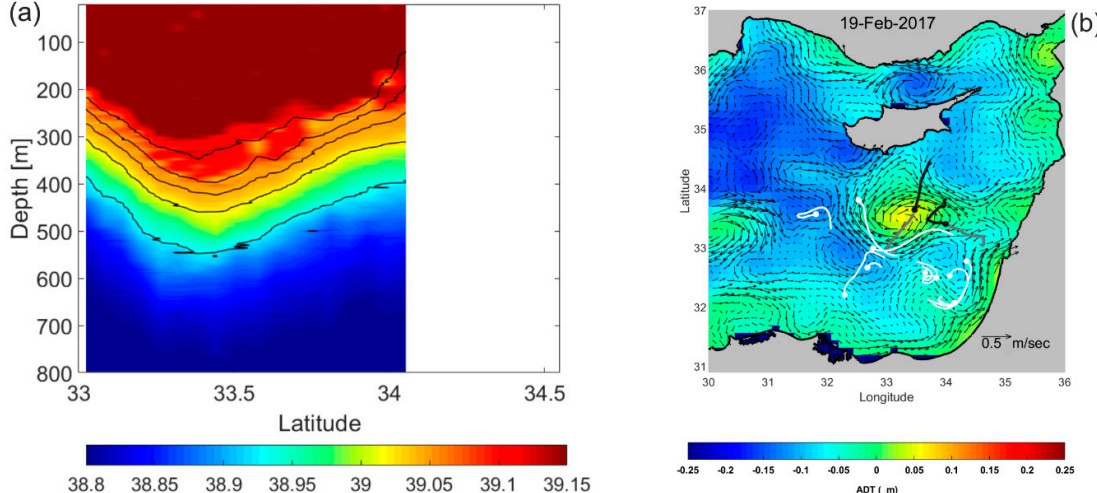

**Figure 14.** Winter salinity field of transect 6 (dates in Table 2) isopycnals (black lines, intervals of 0.1 kg/m³) overlaid (**a**); ADT image of the day when the glider crossed the center of the eddy (**b**). The transect is relative to the glider track depicted to the left, the dot is the position of the glider on the same day of the image, the black line is relative to 4 days before and the gray line is relative to 4 days after. White curves are 4-day trajectory segments of drifters (shown with white dots).

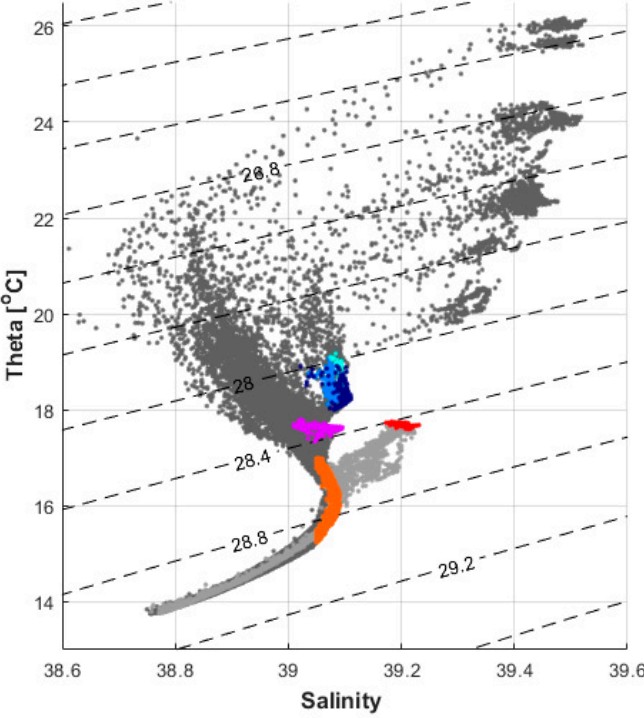

**Figure 15.** Theta-S diagram in the CE region obtained from the fall (winter) glider data in dark gray (light gray), the dashed lines are isopycnals. Turquoise, light blue, dark blue and magenta dots correspond to the high salinity core found in the transects 2, 3, 4 and 5 respectively. Red dots are relative to values between 20–200 m in winter transect, while orange dots correspond to the salinity maximum associated to the LIW, both for fall and winter missions.

To analyze further the thermohaline properties of the high salinity core a Theta-Salinity (TS) diagram of the CE region is depicted in Figure 15. All the glider temperature and salinity data of transects 1 to 6 (see Table 2 and Figures 12 and 14) are represented during fall (dark grey) and winter (in light grey). The high salinity core data, found in transects 2, 3, 4 and 5 are respectively plotted in turquoise, light blue, dark blue and magenta. The red is used to distinguish the homogeneous salty water mass present in the upper layer (20–200 m) inside the CE during winter (Figure 14a), while orange corresponds to the salinity maximum associated with the LIW, in both seasons. All the high salinity cores present the salinity of the LIW, with exception of a few values of the core in transect 5. The density of each core ranges between 28 and 28.34 kg/m$^3$ and the vertical structure inside the core is mainly driven by temperature (as visible from the isopycnals in Figure 12). The core in transect 5 has a lower temperature in comparison with the others resulting in a higher density (28.36–28.4 kg/m$^3$).

## 4. Discussion and Conclusions

The use of satellite altimetry and SST anomaly in concert with drifter data allowed us to analyze the surface evolution of some of the major sub-basin and mesoscale circulation structures in the eastern LS in two different seasons between September 2016 and August 2017. The eastern LS appears very variable over time: the geostrophic position, the size and the shape of most eddies change in a few weeks due to their interaction or their complete disappearance.

The anticyclonic CE is the most persistent feature in the above-mentioned period covering a broad geographical region south of Cyprus. It evolves while changing in shape, merging with other eddies like the NSE during fall 2016. Drifters were captured by it between February and May 2017 allowing a quantitative study of its kinematics. The mean radius of the eddy is about 40–50 km and the maximum speed of about 50 cm/s is reached at 40 km from the eddy center. The CE moves 150 m/day to the southeast. The NSE is present mainly during fall until the beginning of December, when it becomes partially merged with the CE. Then, it disappears until July when it reappears and it moves ½ degree more to the north. The cyclonic SSE in some periods moves far from the coast to the southwest, merging with other cyclones and disappearing from its most probable position for weeks. Other eddies are present in the area like E1, E2 and LE (Figure 3), but their presence and position during the year of the study seems to be very variable and difficult to easily describe.

Besides eddies, the eastern LS is also characterized by a strong upwelling south of Cyprus in the summer months (September 2016 and May–August 2017), and other features occasionally present like the MMJ and the cold front east of Cyprus. The MMJ is present during the whole month of September 2017, crossing the basin from west to east, clearly traced by the cold water in the SST anomaly images, by the drifter tracks and by the geostrophic currents computed from the ADT fields. The jet skims the southern coast of Cyprus and flows toward the Syria/Lebanon coast, where it decreases in strength and it turns north along the coast. The disappearance of the jet corresponds to a wider extension of the CE to the north. The thermal zonal front, spanning from the southeast coast of Cyprus to the Syria/Lebanon coast represents a persistent feature from October to the beginning of March, concomitant with the cooling of the northern part of the basin. The front with a prevailing zonal orientation is first jagged and undefined, then it becomes sharper characterized by a strong current in the warmer side and finally, when the currents weaken, the north intrusions start to degrade the front. The upwelling in front of Israel at the end of November 2016 represents another sporadic event.

The glider campaigns sampled the vertical structures present in the area allowing a general description of a few circulation features in fall 2016 and in winter 2016–2017. The CE is the most sampled eddy during the fall season; it shows the usual thermohaline structure of the area, modulated below the mixed layer (50–100 m), by the downward bending of all the physical parameters measured. However above it, the isopycnals bend upward (Figure 12c,e,g) and in between the up and downward isopycnals, a homogeneous volume of water is present. This kind of eddies has been described as anticyclonic modewater eddies or submesoscale coherent vortices in different basins of the world [46–50]. The CE has a strong signature in ADT, but usually these eddies

are also characterized by a negative SST anomaly, that was not seen at least during fall and winter. In summer, instead, the signature in SST anomaly was evident for three full months, but it was not supported by any glider measurement.

The existence of a high salinity core in the CE area was previously reported in studies implying CTD and XCTS surveys [3,25,30,51–53]. More recently, this lens was also described in glider missions, which took place in late fall and at the beginning of winter during the years 2009, 2010 and 2011–2012 [54]. However, in these earlier glider observations, the maximum of the salinity (around 39.3 PSU) showed higher values and lower temperature (around 17 °C) with respect to the LIW. In the present study, the core has the same salinity and a higher temperature (around 18.5 °C) than the LIW. A minimum in oxygen is evident in the lower part of the core; it is probably related to biological processes as those indicated in the Tropical North Atlantic by [48,55] and in the Mediterranean Sea by [56].

The CE shows a north-south asymmetry with steeper isolines in the north, where the bathymetry is shallower (Figure 13). This can be due to an interaction between the CE and the bathymetry as already shown in previous studies [13,30,36,57]. The shallow bathymetry is probably responsible for the strengthening of the geostrophic currents in the northern part of the CE, also underlined by the ADT maps analyzed in this study and the results described in [4].

Based on the obtained results, it is difficult to identify a specific process involved in the generation of the high salinity core found in the CE region. We can only speculate on the possible generation mechanisms. The shallower bathymetry in the northern part of the CE region, leads to a rising of the high salinity maximum of the LIW during summer or fall. The interaction of this layer with the bathymetry could generate instabilities that induces the creation of a nucleus of high salinity. Another origin of the core thermohaline properties could be related to the water mass created during winter mixing, as mentioned in [25,56]. This is consistent with the results of [58,59], who showed that the CE is able to maintain its core temperature and salinity characteristics for periods of up to two years. However further understanding of the local mechanisms of water mass exchanges and mixing processes is needed. This will be the subject of future studies.

**Author Contributions:** Conceptualization, E.M. and P.-M.P.; Data curation, L.S., R.G. and D.H.; Formal analysis, E.M. and L.S.; Funding acquisition, P.-M.P.; Investigation, D.H. and H.G.; Validation, L.S. and R.G.; Writing—original draft, E.M., L.S.; Writing—review & editing, L.S., R.G. and P.-M.P.

**Funding:** This study is part of the CINEL project sponsored by the U.S. Office of Naval Research (ONR) under grant N000141512459.

**Acknowledgments:** We thank the following persons who have helped with the sea-going operations and the processing of the data: Antonio Bussani, Milena Menna, Piero Zupelli, Massimo Pacciaroni, Stefano Kuchler and Aya Hozumi.

**Conflicts of Interest:** The authors declare no conflict of interest.

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
