# Peer review of "On the Variability of the Circulation and Water Mass Properties in the Eastern Levantine Sea between September 2016–August 2017"

_water, doi:10.3390/w11091741_

Round 1

Reviewer 1 Report

It should be change "back" to "black" on the line 283

Author Response

We would like to thank the reviewer for careful and thorough reading of this manuscript.

Our response follows in italics.

Comments and Suggestions for Authors

It should be change "back" to "black" on the line 283

We changed it.

Reviewer 2 Report

The manuscript reports on measurement campaigns in the eastern Levantine Sea (LS) performed between 9/2016 and 8/2017. The campaigns use a total of 16 drifters and 3 gliders to probe the complex multi-scale circulation patterns in the LS. In addition, satellite sea surface temperature measurements and geostrophic currents derived from satellite altimetry (ADT) measurements are presented to explain the broad structure of the LS circulation and to provide a context for the in-situ measurements. The drifter data show good agreement with the circulation derived from satellite data. Eddies capture several of the drifters and this allows to determine in more detail their spatial properties. The glider missions have been used to probe the Cyprus eddy (CE) and its surroundings under various circumstances, leading to a detailed description of its vertical structure.

General comments:

The manuscript is interesting to read and relevant to the journal/section and the special issue that it was submitted to. The results are presented in a clear and concise manner. In particular, the presentation of measurements alongside a picture of the geostrophic currents works well to guide the reader through the results. My only criticism here would be that the resolution of these figures is rather low, and drifter/glider trajectories are not always easy to distinguish.

Specific comments:

L37: focusing on the LS

L63: below

L122: The procedure to "eliminate obvious spikes" is not immediately clear, is this the procedure described in Ref. [32]?

Fig. 2.: Maybe the drifters could be represented by different colours to make it possible to follow them at least in the less dense areas

L154: "loops in the drifter tracks were identified by defining all the longitudes and latitudes between two longitude maxima [35]" became clear only after looking at Fig. 3 of Ref. [35], please consider adding a few more sentences

L161: L4 maps of SST are a combination of measurements and interpolation, especially in winter (as in Fig. 4) there can be large gaps in the satellite coverage. Have you checked to what extent the data correspond to actual measurements at L3?

Fig. 3.: Red spell-checker lines remain in the figure, also the spell checker is right about the spelling of Lebanon

L305: Rephrase "Anyhow, …"

Fig. 9a: lonongitude -> longitude

L333-L336: I am not fully convinced that with only 3 drifters you can make such a strong claim about the radius of the CE. Wouldn't it be possible for the drifters to change to an orbit further inside or outside without the actual extent of the CE changing?

L356: downward (bending) of the isolines

Fig. 12: Panels f, h and j show multiple glider trajectories without specifying which one is shown in the corresponding transect. Caption: ADT field and currents are on the right, not left.

Fig. 13: It is unclear how 2, 3, and 5 continue after the tracks cross, perhaps they can be numbered also at the opposite end.

Fig. 15: Theta -> Temperature. Explain that the dashed lines represent density in the caption.

Author Response

We would like to thank the reviewer for careful and thorough reading of this manuscript and for the constructive comments and suggestions, which help to improve the quality of this manuscript. Our response follows in italics.

Specific comments:

L37: focusing on the LS                                                         

Changed

L63: below

Done

L122: The procedure to "eliminate obvious spikes" is not immediately clear, is this the procedure described in Ref. [32]?

Yes, we added the reference.

Fig. 2.: Maybe the drifters could be represented by different colours to make it possible to follow them at least in the less dense areas

The figure was redrawn using different colors.

L154: "loops in the drifter tracks were identified by defining all the longitudes and latitudes between two longitude maxima [35]" became clear only after looking at Fig. 3 of Ref. [35], please consider adding a few more sentences

The explanation of the method to identify loops in the drifter tracks has been rephrased.

L161: L4 maps of SST are a combination of measurements and interpolation, especially in winter (as in Fig. 4) there can be large gaps in the satellite coverage. Have you checked to what extent the data correspond to actual measurements at L3?

Yes, we have checked. The images with extended interpolated areas were excluded  and only the images with little interpolation were included and checked. We believe that the shape and evolution (in subsequent images) of the features examined are realistic. Furthermore, most of them have a good match with the drifter tracks.

 Fig. 3.: Red spell-checker lines remain in the figure, also the spell checker is right about the spelling of Lebanon

The red spell-checker lines were removed and Lebanon was corrected.

L305: Rephrase "Anyhow, …"

       Rephrased

Fig. 9a: lonongitude -> longitude

Done

L333-L336: I am not fully convinced that with only 3 drifters you can make such a strong claim about the radius of the CE. Wouldn't it be possible for the drifters to change to an orbit further inside or outside without the actual extent of the CE changing?

The reviewer is right, drifters can be in the eddy at different distances from the center, and their loops can eventually change in time, from one loop to the other, should there be divergence or convergence in the eddy Core.  The scatter plot of drifter mean speed in a loop, versus the loop radius, (see Fig. 9 in report [35]) indicate that maximum velocities occur for a radius of 40-50 km, despite obvious variabilities due to other factors (motion can be elliptic instead of circular, strength of the eddy can vary with time, etc.). The value of 40-50 is in good agreement with satellite altimetry maps. Hence, our results are compatible with the concept of an eddy, which is in solid body rotation in its core (anticyclonic vorticity, less than 40-50 km). For larger radius, the speed decreases with distance from the center and the vorticity becomes cyclonic. We have modified the text accordingly in the revised document.

L356: downward (bending) of the isolines

       Changed

Fig. 12: Panels f, h and j show multiple glider trajectories without specifying which one is shown in the corresponding transect. Caption: ADT field and currents are on the right, not left.

       The caption was modified and the glider tracks were identified as well.

Fig. 13: It is unclear how 2, 3, and 5 continue after the tracks cross, perhaps they can be numbered also at the opposite end.

We added the numbers at the other end.

Fig. 15: Theta -> Temperature. Explain that the dashed lines represent density in the caption.

We changed the captions adding the dashed line explanation.

Reviewer 3 Report

Review of the manuscript: “On the variability of the circulation and water mass properties in the eastern Levantine Sea between September 2016 – August 2017”

General Comment:

I think the manuscript could be a useful contribution in the general discussion on the importance of mesoscale and sub-mesoscale structures in the Levantine Sea. Furthermore, taking advantage of the glider data, it describes the vertical structures of these features.

The multiplatform approach used by the authors is not new, but  allows them to describe water mass characteristics and ocean dynamics at different spatial scales both at surface and at intermediate depths. The description of the water column properties along depth is really interesting, especially when studying sub basin and mesoscale structures whose sampling is usually arduous.

Nevertheless, I have reported a list of comments and suggestions, that should be addressed by the authors before the manuscript become suitable for publication. In my opinion the manuscript could be accepted for publication after major revision.

L19                         How many drifters? Three?

L29                         “the lens” is the high salinity core just described?

L44                         EGYPT seems to be the right acronym

L35-59                  I think a figure could help describing the general circulation of the area. Probably a new version of figure 1a or figure 2 could make this job

L64                         western

L66                         Levantine Deep Water

L72-74                  This approach has been already applied in the Mediterranean Sea and in other areas, but the authors never refer to previous similar experiences. Some references on this argument are indicated.

L83                         Satellite altimetry data

L99                         CTD?

L105-106              why no sampling during downcast was realized?

L112                      Glider velocities seems to be quite low, do the authors considered any implication on the sampling of the larger structures? How long did the glider take to cross the CE?

L119                      Oxygen data are commented in several sections of the manuscript and they are also reported in several figures. Even if the authors do not rely on oxygen absolute values, the relative values of oxygen concentration should be reported in the text and in figure (labels).

L124                      As Fluorescence and backscattering values are discussed in the manuscript, this advice on their quality sounds strange. Do the authors rely on these data or not?

FIGURE 1             colour scale label are not readable

FIGURE 2             Using a different colour for each drifter could help in identifying main structures in the area

L167-170              The authors calculate and show in figures the SST anomaly maps, not the SST. Please correct the text and the figure captions.

L214-215              please indicate the Syrian coast in the figure

L216                      surrounding water

L217-218              The Cyprus upwelling should be probably described before

L231                      Can the authors give an explanation or a hypothesis for this different current pattern?

L235-237              In the previous lines (L209), the authors stated that a strong jet was evident since September 2016, characterized by low temperatures and strong currents. Now they state that the jet was not defined in the temperature data. Please clatify

FIGURE 4             Lines representing the tracks of glider and drifters should be more evident

FIGURE 5             Lines representing the tracks of glider and drifters should be more evident

L302                      a decrease in temperature

L307                      Can the authors give another explanation? An upwelling system should be evident also in deeper isolines (i.e. cold temperature doming)

L336                      due to the not perfectly circular shape of the eddy

L344-350              The authors stated that the very surface layer (0-20 m depth) was not sampled by the gliders, so I think is difficult for them to describe the layer from surface to 40 m depth

FIGURE 11           Please change isoline color and add the labels

L425-418              In the final section of the manuscript, the authors offer an explanation of this asymmetry basing on the effect of bathymetry. Probably this hypothesis should be also described here together with other possible explanations.

L460-461              please see previous comments on the use of fluorescence and backscattering data

L528                      CTD

Many useful references are summarized in the PRE SWOT CRUISE REPORT “Mesoscale and sub-mesoscale vertical exchanges from multi-platform experiments and supporting modeling simulations: anticipating SWOT launch (PRE-SWOT)” DOI: 10.20350/digitalCSIC/8584

Additional useful references:

Aulicino, G., Cotroneo, Y., Ruiz, S., Sanchez Roman, A., Pascual, A., Fusco, G., Tintore, J., and Budillon, G. (2018). Monitoring the Algerian Basin through glider observations, satellite altimetry and numerical simulations along a SARAL/AltiKa track. J. Marine Systems,179:55-71.

Bosse, Anthony, et al. "A submesoscale coherent vortex in the Ligurian Sea: From dynamical barriers to biological implications." Journal of Geophysical Research: Oceans 122.8 (2017): 6196-6217.

Cotroneo Y, Aulicino G, Ruiz S, Pascual A, Budillon G, Fusco G, Tintore J, 2016. Glider and satellite high resolution monitoring of a mesoscale eddy in the Algerian basin: Effects on the mixed layer depth and biochemistry. J. Mar. Syst. doi: 10.1016/j.jmarsys.2015.12.004

Font, J., Isern-Fontanet, J., Salas, J.J., 2004. Tracking a big anticyclonic eddy in the Western Mediterranean Sea. Sci. Mar. 68 (3), 331–342.

Olita, A., Ribotti, A., Sorgente, R., Fazioli, L., Perilli, A., 2011. SLA-chlorophyll-a variability and covariability in the Algero-Proven ̧cal Basin (1997–2007) through combined use of EOF and wavelet analysis of satellite data. Ocean Dyn. 61, 89–102.

Pujol, M.I., Larnicol, G., 2005. Mediterranean Sea eddy kinetic energy variability from 11 years of altimetric data. J. Mar. Syst. 58, 121–142.

Troupin, Charles, et al. "The AlborEX dataset: sampling of sub-mesoscale features in the Alboran Sea." Earth System Science Data 11.1 (2019): 129-145.

Author Response

We would like to thank the reviewer for careful and thorough reading of this manuscript and for the comments and constructive suggestions, which help to improve the quality of this manuscript. Our response follows in italics.

Specific Comment:

L19                         How many drifters? Three?

Yes, we rephrased.

L29                         “the lens” is the high salinity core just described?

       Yes, it is. We corrected the text.

L44                         EGYPT seems to be the right acronym

       OK.

L35-59                  I think a figure could help describing the general circulation of the area. Probably a new version of figure 1a or figure 2 could make this job

We believe that figure 3, showing one year mean ADT and the main sub-mesoscale features superimposed describes the general circulation of the area. The figure 2 and 3 were recreated to better describe the area of the study.

L64                         western

       Changed.

L66                         Levantine Deep Water

       Changed.

L72-74                  This approach has been already applied in the Mediterranean Sea and in other areas, but the authors never refer to previous similar experiences. Some references on this argument are indicated.

The suggested references are included in the text and used in the conclusions.

L83                         Satellite altimetry data

It was changed.

L99                         CTD?

Yes, the GPCTD is a pumped CTD, the acquired data do not need the thermal lag, and it is different from the fluxed CTD.

L105-106              why no sampling during downcast was realized?

This strategy was adopted to save power in long missions. The pumped CTD consumes more power them the regular un-pumped.

L112                      Glider velocities seems to be quite low, do the authors considered any implication on the sampling of the larger structures? How long did the glider take to cross the CE?

The glider speed can be variable, on average is around 20 km/day a big structure like the CE was crossed in about 8 days. We know that the data are not fully synoptic, but SST and ADT confirm the structures understudy vary at longer time scales. 

L119                      Oxygen data are commented in several sections of the manuscript and they are also reported in several figures. Even if the authors do not rely on oxygen absolute values, the relative values of oxygen concentration should be reported in the text and in figure (labels).

We reported the values in fig. 11, but we prefer to keep only the iso-oxygen in fig 12, since the different panels shows data sampled by different sensors on different gliders. An oxygen inter-calibration between gliders was not performed as well as no in situ samples were collected. We believe that oxygen gradient in space is important to be reported, because relevant to a better description of the features.

L124                      As Fluorescence and backscattering values are discussed in the manuscript, this advice on their quality sounds strange. Do the authors rely on these data or not?

It is a best practice to perform a dark count for optical measurements during long missions, at the beginning and at the end of it. This is necessary to correct the drift during the mission, especially if no in situ data are collected. The OGS sensor was calibrated about a year before the beginning of the mission; nevertheless, we believe that the absolute value has to be taken with some care. We did not collect in situ samples to compare with our measurements due to the fact, that no oceanographic vessel was involved in the experiment. We rely on the data and we describe the spatial gradient and the pattern marked also by other parameters.

FIGURE 1             colour scale label are not readable

The figure has now higher resolution and the colorbar is readable.

FIGURE 2             Using a different colour for each drifter could help in identifying main structures in the area

       The figure was changed accordingly to the suggestion.

L167-170              The authors calculate and show in figures the SST anomaly maps, not the SST. Please correct the text and the figure captions.

      We corrected.

L214-215              please indicate the Syrian coast in the figure

      Done in fig2 and fig3.

L216                      surrounding water

      Changed.

L217-218              The Cyprus upwelling should be probably described before

We described the MMJ first and the upwelling right after because the MMJ faded and the upwelling stays longer.

L231                      Can the authors give an explanation or a hypothesis for this different current pattern?

The drifters were released in different location of the CE area, they reached the northern CE ad different times and followed the current characterized special and temporal variability. Some drifters did not followed the ADT maps in a specific period because of the poor precision of ADT along the coast.

L235-237              In the previous lines (L209), the authors stated that a strong jet was evident since September 2016, characterized by low temperatures and strong currents. Now they state that the jet was not defined in the temperature data. Please clarify

In L235-237   we describe the MMJ first. We added to the thermal: ‘northeast of Cyprus’  to better distinguish the two.      

FIGURE 4             Lines representing the tracks of glider and drifters should be more evident

The figure has now higher resolution and the tracks are more evident.

FIGURE 5             Lines representing the tracks of glider and drifters should be more evident

The figure has now higher resolution and the tracks are more evident.

L302                      a decrease in temperature

       Done.

L307                      Can the authors give another explanation? An upwelling system should be evident also in deeper isolines (i.e. cold temperature doming)

The signal in the deep isolines is evident in the coastal area, but the glider sampled an area quite away from it. In our opinion the colder advected waters are probably only in the very surface layer.

L336                      due to the not perfectly circular shape of the eddy

       Done.

L344-350              The authors stated that the very surface layer (0-20 m depth) was not sampled by the gliders, so I think is difficult for them to describe the layer from surface to 40 m depth

True, we changed the text to ‘between 20 m to 40 m’.

FIGURE 11           Please change isoline color and add the labels

The figure has now higher resolution, lines are more evident and we add the labels.

L425-418              In the final section of the manuscript, the authors offer an explanation of this asymmetry basing on the effect of bathymetry. Probably this hypothesis should be also described here together with other possible explanations.

Other authors found the asymmetry in eddies among them Cotroneo et al. 2016 or Chunhua et al. 2018. With the sampling strategy adopted during the CINEL experiment, we are not able to compute the quasi-geostrophic vertical filed or the relative geostrophic vorticity to infer more information about the CE. Our hypothesis states that the shallow bathymetry is probably responsible for the strengthening of the geostrophic currents in the northern part of the eddy, in agreement with other results found in literature, that highlight the strengthening of current as the responsible of the asymmetry of the eddies .

L460-461              please see previous comments on the use of fluorescence and backscattering data

      Please refer to the answer regarding L124.

L528                      CTD

We changed it.

Round 2

Reviewer 3 Report

I find this new version of the manuscript really improved.

I thank the authors for their additional efforts.

The manuscript can now be accepted in its present form.

Best regards.